# 2.5 Million Annual Deaths—Are Neonates in Low- and Middle-Income Countries Too Small to Be Seen? A Bottom-Up Overview on Neonatal Morbi-Mortality

**DOI:** 10.3390/tropicalmed7050064

**Published:** 2022-04-21

**Authors:** Flavia Rosa-Mangeret, Anne-Caroline Benski, Anne Golaz, Persis Z. Zala, Michiko Kyokan, Noémie Wagner, Lulu M. Muhe, Riccardo E. Pfister

**Affiliations:** 1Neonatal Division, Geneva University Hospitals, 1205 Geneva, Switzerland; persis.zala@hcuge.ch (P.Z.Z.); riccardo.pfister@hcuge.ch (R.E.P.); 2Global Health Institute, University of Geneva, 1205 Geneva, Switzerland; michikokyokan@gmail.com; 3Faculty of Medicine, University of Geneva, 1205 Geneva, Switzerland; 4Obstetrics Division, Geneva University Hospitals, 1205 Geneva, Switzerland; anne-caroline.benski@hcuge.ch; 5Harvard T.H. Chan School of Public Health, Boston, MA 02115, USA; 6Center for Education and Research in Humanitarian Action, Faculty of Medicine, University of Geneva, 1205 Geneva, Switzerland; anne.golaz@unige.ch; 7Centre Medico-Chirurgical-Pédiatrique Persis, Ouahigouya BP267, Burkina Faso; 8Pediatric Infectious Diseases Division, Geneva University Hospitals, 1205 Geneva, Switzerland; noemie.wagner@hcuge.ch; 9College of Health Sciences, Addis Ababa University, Addis Ababa 1000, Ethiopia; muhe1952@gmail.com

**Keywords:** neonatal mortality, neonatal morbidity, neonatal rights

## Abstract

(1) Background: Every year, 2.5 million neonates die, mostly in low- and middle-income countries (LMIC), in total disregard of their fundamental human rights. Many of these deaths are preventable. For decades, the leading causes of neonatal mortality (prematurity, perinatal hypoxia, and infection) have been known, so why does neonatal mortality fail to diminish effectively? A bottom-up understanding of neonatal morbi-mortality and neonatal rights is essential to achieve adequate progress, and so is increased visibility. (2) Methods: We performed an overview on the leading causes of neonatal morbi-mortality and analyzed the key interventions to reduce it with a bottom-up approach: from the clinician in the field to the policy maker. (3) Results and Conclusions: Overall, more than half of neonatal deaths in LMIC are avoidable through established and well-known cost-effective interventions, good quality antenatal and intrapartum care, neonatal resuscitation, thermal care, nasal CPAP, infection control and prevention, and antibiotic stewardship. Implementing these requires education and training, particularly at the bottom of the healthcare pyramid, and advocacy at the highest levels of government for health policies supporting better newborn care. Moreover, to plan and follow interventions, better-quality data are paramount. For healthcare developments and improvement, neonates must be acknowledged as humans entitled to rights and freedoms, as stipulated by international law. Most importantly, they deserve more respectful care.

## 1. Introduction

Neglected diseases still kill and disable too many, but no age group has in this regard received less attention than newborns. A newborn or neonate is a human from age 0 to 28 days. Worldwide, their contribution to death and disease is overwhelming and unequaled by any condition, while at the same time, its prevention and treatments are simple, relatively cheap, and efficient. While rich countries have very reduced neonatal mortality, the poorest regions and conflict zones suffer the highest losses [1]. Today, these figures are expected to have worsened with the recent COVID-19 pandemics bringing a shortage of basic care [2] and equipment, as well as twice as likely preterm delivery of infected pregnant women [3,4].

### 1.1. Definition of Live-Birth

In 1950, mainly for public health and statistical purposes, the WHO defined the term “live birth” as any human wholly extracted from the mother, whatever the gestational age, showing any sign of life, such as voluntary movement, heartbeat, or pulsation of the umbilical cord, for however brief a time [5]. The WHO recommends a declaration of birth, dead or alive, from 22 weeks post-menstrual age. Although this definition appears clear in theory, in clinical practice, cultural, environmental, and emotional factors and ignorance lead to uncertainty even in high-income countries, let alone where no skilled birth attendant is available. For instance, many low- and middle-income countries (LMIC) lack precise post-menstrual age information and skilled birth attendant. Even high-income countries are not spared from these limitations to a certain extent.

### 1.2. Human Rights of the Newborn

The bases for neonatal rights were laid in 1948 when the Universal Declaration of Human Rights proclaimed the inalienable rights to which everyone is entitled as a human being [6]; everyone including every newborn. Much later, the Convention on the Rights of the Child (CRC), adopted by the United Nations General Assembly in 1989, ensured that children’s health became a human rights issue [7]. According to Article 24 of the CRC, all children have a right to the highest attainable standard of health and health care, and the member states have an obligation to reduce child mortality.

The United Nation’s Committee on the Rights of the Child has issued several resolutions specifying newborns as part of the child’s right to health. For example, General Comment No. 15 reinforced the legal obligations of states to reduce child mortality and “urged particular attention to neonatal mortality, which constitutes an increasing proportion of under-5 mortality” [8]. Other fundamental documents supporting newborn rights include the Parma Charter on the Rights of the Newborn (2011), which provides a list of rights related to the promotion and protection of newborn health [9], and the Abu Dhabi Declaration for Women and Children (2015), which calls for a more strategic focus on reproductive, maternal, newborn, child, and adolescent (RMNCA) health in humanitarian and fragile settings [10].

These international documents of law recognize that newborns have fundamental rights and freedoms, including the rights to survival, health, development, legal identity from birth, protection from harm, violence, and neglect, and a caring, loving, and nurturing environment everywhere, even in humanitarian and fragile settings. Yet, despite this international legal framework being in existence since several decades now, when it comes to weighing up the choices based on moral, emotional, religious, and financial investment for life-and-death decisions, adult and child life is still globally higher valued than newborn life [11].

### 1.3. Mortality—The Tip of the Iceberg

Of the 140 million live births globally, around 2.5 (2.2–2.7) million neonates die [1,12,13]. Notwithstanding the efforts of the past 20 years with the Millennium Development Goals and Sustainable Development Goals, the neonatal period remains the most likely period for a child to die. Global neonatal mortality accounts for nearly half of the under-5-year mortality and occurs 98% in LMIC [1,13]. In addition, due to the very high immediate postnatal mortality, the risk of underreporting remains major [14,15]. Many newborns might die before being given a name or being registered as a live birth. Accounting for both stillbirths and neonatal deaths is estimated to double the 2.5 million deaths [15,16].

In 2019, sub-Saharan Africa and Southern Asia alone accounted for 80% of the world’s neonatal deaths. Two-thirds of these occurred on the first day of life [13] and almost three-quarters within the first week. The place of birth, thus, becomes the first survival challenge faced by a neonate with a risk of death up to 20 times higher in sub-Saharan Africa and Southern Asia [1] and higher for newborns during conflicts and humanitarian emergencies, where they receive even lower priority [17]. Neonatal mortality is so high that it has become the “new normality”. It profoundly affects social and family structure and women’s health. Maternal risks increase with the repeated pregnancies necessary for living offspring and meanwhile women suffer and are stigmatized for each child death. Indeed, traditionally delayed naming most likely mirrors cultural adaptation to high neonatal mortality in an attempt to minimize attachment and suffering in case of early death [18,19].

Close to 80% of all neonatal deaths are due to three leading causes: (1) prematurity and low birth weight, (2) perinatal complications and asphyxia, and (3) sepsis and infection [20,21,22,23,24] with variations according to the region and neonatal period [25]. Prematurity seems to be the most significant contributor to neonatal mortality [13], but there are other causes of mortality within this age group. A large prospective cohort based on verbal autopsy in eight countries in sub-Saharan Africa and South Asia found that birth asphyxia and sepsis accounted for around 70% of neonatal deaths and that preterm birth complications might be the third cause of neonatal mortality, possibly due to incorrect assignment for the death of preterms that die from other causes [26]. Increased causal detail of data within these interlinked domains remains an urgent need.

Mortality can be assumed to be just the tip of the iceberg in the diseases that kill, but more often will lead survivors into chronic disorders and disability. Underestimating and neglecting neonatal disease carries this burden over into adulthood, resulting in considerable societal costs. Neglected neonatal disorders are considered the leading cause of disability-adjusted life years (DALY) in all age groups globally [27]. In LMIC, there is a lack of good quality and appropriately detailed data concerning the perinatal population at all levels (antenatal, birth, mortality, morbidity). When data is not just limited to mortality, it is mostly non-standardized, collected retrospectively in reference centers, and extrapolated to larger populations, thus reflecting poorly intra- and inter-regional variabilities and missing the poorest and least accessible populations [15,28,29]. Comprehensive, accurate, and reliable data with geographic distribution and including quality of care indicators are fundamental to alleviate bottlenecks with effective intervention strategies [13,30,31,32,33,34] and to follow them up.

### 1.4. Need for Action

The world has made substantial progress in child survival since 1990, with the number of neonatal deaths also declining from 5.0 million in 1990 to 2.3–2.7 million in 2019 [13]. However, this decline was considerably slower than that of post-neonatal under-5 mortality, increasing the share of neonatal mortality from 40% to 47% [1]. In 2014, at the end of the MDG era, a Lancet series of 5 articles advocated for quality care at birth and actions to improve neonatal health, urging to invest in neonatal care [19,35,36,37,38], culminating in the ‘Every Newborn Action Plan’ [39]. In 2015, the world committed to the Sustainable Development Goals (SDGs) with the third goal (SDG 3) aiming “to ensure healthy lives and promote wellbeing for all at all ages” [40]. Target 3.2 of these SDGs explicitly focuses on “ending preventable deaths of newborns by 2030, with all countries aiming to reduce neonatal mortality below 12 per 1000 live births”.

Unfortunately, at the current pace, more than 60 countries will not meet this target, with estimated 28 million neonatal deaths between 2020 and 2030 [1]. Action is needed and, without knowledge and visibility on specific neonatal mortality and morbidity, as well as lobbying for their survival and targeted quality care, success can hardly be expected.

## 2. Materials and Methods

We propose an overview by specialists in neonatal care with a large practical experience in LMIC within their domain. The rationale is a bottom-up comprehensive approach from the clinician to the policy maker with two main objectives:
To summarize the main bottlenecks leading to neonatal morbi-mortality, focusing on low- and middle-income countries (LMIC),To link these leading causes of neonatal morbi-mortality to neonatal physiology and clinical challenges, in order to propose interventions that work and are feasible in the field.

We also expect to pinpoint how and why interventions should be implemented in a sustainable fashion, highlighting patient (newborn) respect and rights.

To prepare the review, F.R.-M. and R.E.P. elaborated a synopsis containing the paper structure and the focus of the work, which they then shared with the other authors that gave their input in two rounds. Each author worked independently on their review domain and prepared, following the synopsis, a free review of the attributed domain of expertise. At least two authors worked together on each section as mentioned in the “author’s contributions”. These were then shared, completed, and reviewed by F.R.-M. and R.E.P. F.R.-M. produced then the introduction, discussion, conclusions, and abstract and the completed work that was reviewed by R.E.P. and all authors in several rounds until all authors approved the final manuscript’s version.

## 3. Results

### 3.1. Perinatal and Birth-Related Complications

#### 3.1.1. Definition and Disease Classification

The distinction between ‘live born’ and ‘stillbirth’ is essential to understanding perinatal mortality. According to WHO, a stillbirth is “a baby who dies after 28 weeks of pregnancy, but before or during birth” [41]. Within the stillbirth group, intrapartum-related stillbirths or “fresh stillbirth” are “neonates that show no signs of life at delivery and weigh more than 500 g or are greater than 22 weeks of gestation with intact skin and no signs of disintegration in utero” [42]. Their death is assumed to have occurred within 12 h before delivery, likely from a hypoxic event. These latter deaths alone are estimated at 1.3 million [43,44].

A similarly high number of 1.2 million postnatal deaths originate during labor, i.e., are intrapartum-related, most frequently due to birth asphyxia. Birth asphyxia is defined by a perinatal hypoxic event followed by encephalopathy. The first condition for the definition in LMIC is usually failure to initiate or sustain spontaneous breathing at birth, specified mainly by a persistently low 1 and 5 min Apgar score [43,45]. High-resource settings use preferentially cord blood acidosis below pH 7.0, but this is usually unavailable in low-resource settings [43]. Intrapartum-related neonatal pathologies are also associated with significant morbidity, resulting in an estimated burden of 42 million disability-adjusted life years (DAYLS) [46].

Clinical factors such as antenatal obstetric complications, parity, multiple births, gestational age <37 or >41 weeks, low birth weight, premature rupture of membranes, prolonged labor, and fetal distress are known risk factors of birth asphyxia [47,48,49]. Some of these factors concern antenatal and obstetric issues, others neonatal care, but all need improved coordination between obstetric and neonatal healthcare professionals. Evidence is available that it is possible to improve survival and health of neonates and reduce stillbirths. This is achieved by increasing the coverage of quality antenatal care, with skilled intrapartum care and postnatal care most notably in the areas of neonatal resuscitation and care of the low birth weight and sick newborns [50,51,52].

#### 3.1.2. Antenatal Care: Pregnancy and Delivery

A comprehensive antenatal care program involves a coordinated approach, continuous risk assessment and psychological support that optimally is initiated before pregnancy and extends throughout the postpartum and interpregnancy periods [53]. It should integrate health education and community engagement and the concept of family-centered care [54]. A 2020 Cochrane Library review on antenatal interventions for preventing stillbirth, fetal loss, and perinatal death showed clear evidence that reduced antenatal care visits were associated with increased perinatal mortality [55]. However, what works in high-income countries, may not work in LMIC. For example, where the emphasis in programs of high-income countries focusses on ‘client’ autonomy for decision making during pregnancy, labor, or the postnatal period; the emphasis in LMIC may be more on enabling health systems that promote social justice, respect, and access to antenatal and postnatal health care, including delivery in a healthcare facility [56]. In high-income countries where women receive good quality skilled intrapartum care, the proportion of stillbirths is less than 10% of all births [57].

In LMIC, a significant proportion of women give birth at home. Worldwide, some 34% of deliveries, or 45 million births, occur without a skilled birth attendant [58]. This situation is commonplace in sub-Saharan Africa and Southern Asia [58]. Moreover, access to skilled birth care and emergency obstetric care is lowest among the poor, who therefore, suffer the greatest brunt of maternal and neonatal mortality and morbidity related to childbirth [58].

In the 1990s, the components of emergency obstetric and newborn care (EmONC) were determined by WHO, UNICEF, and UNFPA to achieve MDG [59]. In summary, the EmONC interventions recommend that all providers become capable of managing these common complications to decrease the need for referral and improve outcomes [60]. The basic package—the BEmONC, includes antibiotics, anticonvulsants, uterotonics, manual vacuum aspiration, vacuum-assisted delivery, manual removal of placenta, and newborn resuscitation. These interventions of BEmONC are completed by cesarean section and blood transfusion for a Comprehensive package, CEmONC. Quality training in EmONC goes beyond bringing together providers for classroom and clinical practice for some days; it needs to be competence-based and quality-focused, thus providing post-training follow-up and integrating clinical learning interventions [61].

There is a remarkable lack of good quality data on the impact of providing the emergency obstetric care package of care on stillbirths or perinatal mortality, and randomized controlled trials would be unethical. Most studies are observational, but Yakoob et al. in their review used a Delphi consultation to evaluate the effectiveness of EmONC and CEmONC on perinatal outcomes, including stillbirth, suggested that EmONC had the potential to avert 45% of intrapartum stillbirths and CEmONC up to 75% and a reduction in stillbirths by 45% through skilled birth attendance [58]. Another observational study from Sudan reported a 25% reduction in stillbirths and neonatal deaths through training of village midwives compared to controls [62]. Similar results with trained midwives were reported in Bangladesh with a 24% reduction in stillbirth rate after introducing a safe motherhood program and promoting skilled birth attendance [63].

Recent studies confirmed an improved outcome for deliveries occurring in healthcare facilities during the past decades. However, they did not identify an increase in quality of care during delivery and the postpartum period [64] despite clear evidence that quality care during delivery reduces early neonatal mortality [65]. A 2020 Cochrane review found that while many antenatal interventions did not reduce stillbirths or perinatal deaths, some interventions were beneficial, such as balanced maternal energy/protein supplements, midwife-led models of care, skilled and trained traditional birth attendants, fetal heart monitoring during delivery, and delayed newborn bathing [55]. Even if continuous electronic fetal heart rate might be unavailable in LMIC, intermittent fetal auscultation may still be effective and easily implemented.

#### 3.1.3. Postnatal Care and Neonatal Resuscitation

After birth, the effect of some degree of hypoxia can be reversed, reduced, or avoided. Neonates are remarkably resilient to hypoxia due to their fetal hemoglobin and organ function at low oxygen levels in utero. In addition, and in contrast to adults, their vital organs, particularly the heart, are almost always healthy. However, as the birth process presents additional hypoxic stress, about one in ten neonates, that is, 10–13 million neonates every year, need some form of neonatal resuscitation at birth [66]. Neonatal resuscitation focuses on oxygenation of the lungs to rapidly prime the healthy heart with oxygen to further support adaptational processes.

Without immediate and fast intervention, though, hypoxia will lead to handicap and death. Over 99% of interventions are efficient within one minute, for this reason called the ‘Golden Minute’ [67,68]. The target period is concise, indications are straightforward, based exclusively on heart rate and breathing, and the effectiveness is swift and long-lasting. Neonatal resuscitation has been considered one of the most cost-effective medical interventions [69,70]. Unfortunately, immediate newborn intervention is often missed at birth, as the focus remains on the delivering mother. Delaying initiation of neonatal resuscitation increases the complexity of resuscitation and neonatal risks [71,72]. Essential neonatal resuscitation equipment is cheap and re-usable, composed of a self-inflating ventilation bag-and-mask and a manual suctioning device. When necessary, initial ventilation is recommended with room air and oxygen is rarely needed. The most complex technical element is an efficient thermal source, usually a radiant heater. It is not expensive per se but less transportable and depends on a solid electric grid. However, heat loss prevention by fast drying and wrapping and using the maternal body as a heat source is effective and easily instigated, at least in full-term neonates. In a survey of 98 health care facilities in Ethiopia, only 27% had an available heat source in their delivery room, 12% did not assess breathing at birth, and only 66% had the recommended low-cost, essential equipment for neonatal resuscitation [73].

The most critical element for adequate neonatal resuscitation is the intervention without delay by a skilled person [67,68,74]. The manual skills are simple to learn, and the algorithm to follow basic. Lee et al. estimated in their meta-analysis that from the 130 million neonates born, 10 million (5–10%) require simple stimulation with drying and rubbing, 6 million (3–6%) basic resuscitation with bag-and-mask ventilation, and just under 1 million (<1%) advanced resuscitation [75]. These numbers are close to those we published in Switzerland [66]. Considering that around one million neonates die on their first day of life from birth-related complications, efficient neonatal resuscitation at birth appears as one of the most urgent implementations to seek. Lee et al. estimated 30% of intrapartum-related deaths at full-term were preventable, a number possibly lower in preterms and at the community level [75].

Reliable data on these critical minutes after birth are hard to come by. In 1952 Virginia Apgar invented the famous five-element score to be performed at 5 min as “a basis for discussion and comparison of the results of obstetric practices, types of maternal pain relief and the effects of resuscitation” [76]. Subsequently realizing the importance of early intervention, she advocated the same score at 1-min, precisely to prompt early neonatal evaluation and resuscitation [76,77]. It’s well known that the Apgar score alone does not make the diagnosis of birth asphyxia but low scores at five minutes are strong predictors of early neonatal death [78], thus extremely clinically relevant. Furthermore, the score can provide information about newborn health and quality obstetrical and early neonatal care, in the continuum of care, even if they seem to have limited external validity [79].

Twenty-four years after the Apgar score, B. and M. Sarnat published an additional score to describe neonatal encephalopathy in patients with a ‘well-defined’ sign of fetal distress, that is, an Apgar score of ≤5 at 5 min [80]. Together, the Apgar and Sarnat scores have shown to be predictive for the outcome of a hypoxic perinatal event [81], and would be of particular interest where pH determinations and radiological explorations remain inaccessibly expensive.

Successful resuscitation needs follow-up monitoring and usually referral into a facility of higher level, but neonatal transportation demands a significant tribute to neonatal death even with optimal equipment [82], and parents’ travel time distance carry high ‘hidden’ costs. Good evidence shows, however, that centralized deliveries reduce perinatal risks [56]. To that, referral hospitals need a good reputation for patient respect, competence, infrastructure, and results to attract women for delivering. Low-risk deliveries in high-competency healthcare facilities also favor a virtuous reputation cycle.

Hypoxia may be reduced or prevented by rapid adequate artificial ventilation, and a precise diagnosis of hypoxic encephalopathy may allow allocation of resources and possibly brain sparing interventions. Controlled hypothermia for hypoxia-associated encephalopathy is currently the only recommended intervention for neuroprotection in high-income countries [83]. However, its efficiency is less certain in LMIC. A recent large randomized controlled multicentric trial (HELIX trial) found an unexpected increase in the death of asphyxiated neonates receiving controlled cooling. The specific reasons for this difference are not known, but part may be related to associated infections [84]. The best intervention remain prevention through skilled birth and expert resuscitation. Once the hypoxic insult is established, lines of evidence suggest that at least hyperthermia should be avoided as it worsens brain insult [81].

### 3.2. LBW and Prematurity

#### 3.2.1. Definition and Disease Classification

The WHO defines prematurity as birth before 37 completed weeks from the first day of the last menstrual period [85], and is further classified into late and moderate preterm (32 to <37 weeks), very preterm (28 to <32 weeks), and extremely preterm (<28 weeks) [85,86]. Half of the newborns below 32 weeks die in LMIC, whereas, in high-income countries, most survive [86].

Historically, prematurity-related complications have been summarized under “prematurity” as one single entity of under-5 and neonatal mortality [20,87,88,89]. However, prematurity-related deaths are either specific to or concurrent with prematurity, such as congenital anomalies, asphyxia, or sepsis, and may similarly cause death in term infants. Death causes specific to preterms include respiratory distress syndrome (RDS), necrotizing enterocolitis (NEC), and intraventricular hemorrhage [23]. Despite the WHO ICD-10 requesting clinicians not to retain prematurity as the main disease unless no other condition is known [5], there still is a common propensity to assign prematurity as the cause of death without actively searching for the primary cause [90].

An estimated 15 million infants are born preterm every year, and this number is increasing [91,92]. Prematurity associated deaths are considered the leading cause of under-5 mortality [86,90,91]. Over one third of the 2.5 million neonatal deaths are prematurity-related [89]. Little progress has been made in avoiding preterm delivery in high- and low-income settings.

#### 3.2.2. Relation between Prematurity and Low Birth Weight (LBW)

Despite its clear definition, prematurity may be more complex to determine in clinical practice as it requires a precise gestational age. Clinical methods such as the last menstrual period or the New Ballard and Dubowitz scores used for the best estimate of the gestational age, still have an uncertainty of several weeks [93]. More accurate and precise, with only a couple of days of uncertainty, is a pregnancy ultrasound before in the first trimester of gestation. This exam is, however, rarely accessible in low-resource settings.

Weight is universally available, although reported birthweight may have been taken late, up to several days after birth, mainly when registration occurs late. A neonate born at less than 2.5 kg is considered low birth weight (LBW). Preterm infants are generally LBW, but a full-term baby with intrauterine growth restriction can also be LBW [94]. Differentiation from prematurity by birth weight alone is impossible, but LBW remains a surrogate for prematurity when accurate gestational age assessment is impossible. The combined prevalence of low birth weight and prematurity is in the range of 10.2% to 14.6% of all live births [95].

LBW neonates are at higher risk of mortality, stunting, poor neurodevelopment, and adult-onset diseases [20,87,91]. More than 20 million neonates are born LBW every year, and about 70% of global neonatal mortality is within this weight group, again, mainly in LMIC [96,97]. In 2015, three-quarters of the world’s LBW neonates were born in South Asia (47%) and sub-Saharan Africa (25%) [94]. Mortality rates and their causes vary, but comparisons are intricate due to differing gestational age groups, social contexts, and lack of comparable underlying morbidity data. Nevertheless, some risk factors within prematurity and LBW dominate irrespective of the final cause of death; above all, prolonged preterm rupture of membranes and medically indicated preterm delivery [98].

#### 3.2.3. Short- and Long-Term Consequences of Preterm Birth

Preterm infants are prone to complications, particularly intracranial hemorrhage, respiratory distress syndrome, chronic lung disease, intestinal injury, compromised immune systems, and cardiocirculatory disorders [99]. The birth of a preterm infant also brings considerable emotional and economic distress to families and non-negligible implications for public-sector services, such as health insurance, education, and other social support systems [100,101].

A systematic review in 2015 revealed a consistent inverse association between gestational age at birth and economic costs, regardless of the date or country of publication, study design, follow-up period, age of assessment, or cost calculations. This study underpinned the lack of evidence on non-healthcare costs [100]. They estimated for high-income countries the hospitalization costs for extremely preterm 24-week neonates in the range of USD 111,152 to USD 576,972, and for full term neonates of USD 930 to USD 7114 [100]. A Canadian study that analyzed the impact of prematurity on morbidity, mortality, healthcare utilization, and costs [101] determined that the highest national burden was associated with moderate prematurity due to the substantial cost per infant and the large size of this population. The highest individual-level burden was without surprise for early preterm infants [101]. There is no comparable populational data for LMIC. However, no doubt, as moderate and late preterms largely predominate and will survive more, resources requirements will need to increase too [99]. Although the highest medical costs incur during the neonatal period, greater resource utilization and costs extend into childhood [101]. It must be considered though, that in terms of quality-adjusted life year (QALY), due to their long life expectancy, premature infants are, as a matter of fact, less costly per QALY than most severe adult diseases, or even plain hospitalizations for old age [102].

#### 3.2.4. Specific Causes of Death in Preterm Infants

Despite quality perinatal management in prematurity being very effective, even strongly evidence-based interventions often have low coverage and poor quality in LMIC [103,104]. The WHO estimated nearly 18 deaths per 1000 live births in 2016 associated with prematurity. In comparison, the current SDG aims to achieve a reduction in the global neonatal mortality to 12 per 1000 live births, below the present prematurity mortality rate.

Ethiopia’s most extensive observational study investigated specific causes of death and associated factors in more than 1000 deceased preterms primarily based on autopsy and an international expert review of the prospectively collected clinical and laboratory data [94]. In this preterm cohort, the primary cause of death was RDS in 45%, infections combining sepsis, meningitis, and pneumonia in 30%, and asphyxia in 14%. Although this work reports the single most likely cause of death, generally, more than one contributory cause was identified by experts. Hypothermia was the most common contributory cause of mortality, present in 69% of all deaths [94].

#### 3.2.5. Respiratory Distress Syndrome

Effective respiratory function is essential for the high adaptational, and thermoregulation needs immediately after birth. The lungs are also radically changing from fetal liquid to the neonatal air-filled function. It is no surprise that RDS is the single, most important early morbidity in premature neonates and the first cause of mortality in this group [94] whose lungs are most immature and metabolic needs highest. More than 50% of neonates born before 31 weeks of gestation develop RDS [105]. Effective prevention and management strategies such as antenatal steroids, thermoregulation, surfactant therapy, and ventilatory strategies can potentially avoid complications and death in 45–70% [106,107]. Yet, antenatal corticosteroids were used only in 31.2% of mothers of preterms 24 to 34 weeks in Ethiopia [94], hypothermia remains highly prevalent in LMIC [108,109], and blended oxygen and CPAP devices necessary for the most basic ventilation strategies are generally unavailable [110] or largely insufficient in most LMIC [94,111].

#### 3.2.6. Hypothermia

A core temperature of 36.5–37.5 °C is considered normal for newborns and vital for adaptation at birth and survival thereafter. Born from the warm womb into a hostile, cold environment, the newborn struggles with cold stress [112]. Indeed, the naked and wet neonate at birth would need ambient temperatures uncomfortably high for adults, to remain in thermal balance, even more so LBW neonates [113].

In physiological terms, cold stress occurs when the environmental temperature is below the critical temperature for thermal neutrality [114]. It is an urgent condition that triggers the neonate’s efforts to reduce heat loss by vasoconstriction, followed by metabolic heat production, that is, excess energy and oxygen consumption, rapidly leading to its exhaustion. Especially among LBW or premature neonates, metabolic heat production competes with the primary energy and oxygen needs for the body’s essential functions, growth and also fighting illnesses such as RDS [115]. Unfortunately, cold stress has no agreed definition, even though it is largely used in thermal care. The WHO defines cold stress as an axillary or rectal temperature between 36.0–36.4 °C and uses the term interchanging with ‘mild hypothermia’ [112]. However, in physiological terms, even mild hypothermia is already proof of metabolic failure to prevent it. There is no internationally agreed definition of hypothermia in neonatal care [109,116,117,118]. For many specialists, the WHO classification of neonatal hypothermia from 36.0 °C to 36.4 °C as mild, from 32.0 °C to 35.9 °C as moderate, and below 32.0 °C as severe [112] does not sufficiently relate to excess mortality with decreasing temperature [108,119,120].

Unclear definitions, low level of understanding, and trivialization of hypothermia, for instance by considering mild hypothermia equivalent to cold stress, maintain hypothermia as arguably the first and foremost contributor to neonatal death. A study on hypothermia in Ethiopian NICUs demonstrated that nearly 80% of preterm neonates were hypothermic at admission, and many remained hypothermic throughout hospitalization. Lower gestational age and lower birth weight were associated with a higher rate and more severe hypothermia. In a clear dose–response relationship, mortality was significantly associated with lower body temperatures [109]. The study showed that asphyxia, RDS, and resuscitation requirements at birth were also significantly associated with hypothermia.

The WHO warm-chain guides from 1997 rightly attempted to address essential thermal care of the newborn [112]. However, considering even the most recent reports, they failed in reducing hypothermia and hypothermia-associated death [109]. Whether this is due to inconsistent application or inherent flaws, hypothermia prevention and treatment need to be urgently addressed to reduce neonatal mortality, as countermeasures are straightforward and affordable.

#### 3.2.7. Hypoglycemia

Due to their high lean body mass and significant growth, neonates have a high metabolic rate requiring increased energy and oxygen compared to infants and adults [121] Mature neonates have brown adipose tissue [122] with a high thermogenic effect for adaptive non-shivering thermogenesis [113,114], more than doubling heat production in response to cold stress, provid oxygen supply is sufficient [123]. Preterm neonates lack brown adipose tissue that builds up in late gestation. They attempt to satisfy high energetic needs by using glucose from still low glycogen and even structural protein stores [124], frequently resulting in depletion and hypoglycemia. Adding to this, immaturity of enzymatic activity may limit metabolic defense processes for several weeks [125,126,127]. Undiagnosed and untreated, hypoglycemia causes brain damage and may lead to death.

Early maternal contact and breastfeeding are recommended and highly promoted, including by the WHO, but may be insufficient for preterm and LBW neonates who not only lack energy storage but have limited feeding capacity due to immature suckling/swallowing coordination. To worsen the problem, it is common practice to delay enteral feeding, particularly in preterm and growth-restricted LBW neonates, due to a potential (low) risk of necrotizing enterocolitis (NEC). For instance, in the previously cited large Ethiopian study [94], many preterm neonates remained on exclusive maintenance fluids for prolonged periods (expert communication). In the absence of proper parenteral nutrition and blood glucose measurements, it may be reasonably suspected that numerous unrecognized hypoglycemic episodes occur. Moreover, this practice leads to chronic nutritional deficiencies, prolonged hospitalization, and increased mortality. Therefore, improved nutritional care of preterm and LBW neonates should be identified as one of the top priorities to reduce neonatal morbidity and mortality.

#### 3.2.8. Hyperbilirubinemia

Immaturity of the physiologic red blood cell and hemoglobin recycling mechanisms frequently leads to increased serum bilirubin, a particular condition of the newborn. Liver immaturity and increased hemoglobin breakdown almost exclusively increase indirect bilirubin. Worldwide, hyperbilirubinemia occurs in close to 60% of term and 80% of preterm newborns [128,129]. The fat-soluble indirect bilirubin becomes neuro-toxic and eventually causes cerebral damage depending on levels and specific risk factors. Prevention of neurotoxicity is mostly simple and low-cost using phototherapy with blue light at 465 nm wavelength. However, the high costs of serial bilirubin determinations often prevent proper diagnosis, treatment, and follow-up, making unrecognized and untreated hyperbilirubinemia a leading cause of avoidable handicap in LMIC [130,131,132].

### 3.3. Sepsis and Infection

#### 3.3.1. Definitions and Disease Classification

Sepsis is a preventable, potentially life-threatening condition. It is a significant contributor to global mortality, particularly neonatal, and is recognized as a priority by WHO [133,134]. Unfortunately, there is no consensus on the definition to unequivocally differentiate (generalized) neonatal sepsis from (localized) neonatal infection, complicating epidemiological evaluations [135,136]. Despite this lack of consensus, two main categories of neonatal sepsis are widely accepted: early-onset sepsis (EOS) defined as occurring in the first 72 h of life, hence representing fetal-maternal infection; and late-onset sepsis (LOS), which occurs between 72 h and 28 days [136,137]. LOS can be hospital-acquired (HA-LOS) or community-acquired (CA-LOS), an essential difference when considering etiology, treatment, and outcome [138]. CA-LOS appears to be the main form of sepsis in LMIC, but quality data is hard to come by in the community setting [138], likely leading to underestimating its real burden. Furthermore, quality data on incidence and mortality due to neonatal sepsis from most countries worldwide are still lacking [139,140].

Globally, neonatal sepsis is estimated to affect 1.3 to 3.9 million neonates and to account for 400,000 to 900,000 annual deaths [140,141], of which an estimated 84% are preventable [140]. The highest neonatal sepsis incidence is in LBW and premature neonates [136] of LMIC. Preterms that are LBW have a 3–10 times higher risk of sepsis than full-term neonates [136,140]. Maternal chorioamnionitis is the main maternal risk factor [136].

Besides its acute risks, neonatal sepsis is a considerable individual and social burden due to life-long disability and high health costs. In a recent meta-analysis, neonatal sepsis increased the length of hospital stay and hospitalization costs considerably, and the risk of neurodevelopmental impairment (OR: 1.4 to 4.8) [142]. Moreover, neonatal sepsis significantly worsens newborn outcome even in high-income countries. For example, in a large Swiss cohort of preterms below 32 weeks, neonatal sepsis increased the risk of developing cerebral palsy (OR: 3.2) and neurodevelopmental impairment (OR: 1.7) [143]. Nevertheless, the global neurologic burden of neonatal sepsis remains unquantified [137,140].

#### 3.3.2. Etiology

Neonates have a higher risk of sepsis than adults and children due to their immature immunity, potential intrauterine exposure to infection, and in preterms, altered skin and mucosal barriers [142]. Despite the high sepsis incidence in LMIC, quality ecological data on germs remain scarce, though pathogens appear to differ significantly from the flora of high-income countries. In the past decade, in LMIC, Enterobacteria (*Klebsiella* spp. and *E. coli*) and *Staphylococcus aureus* were predominant, while *Streptococcus agalactiae* were rarely reported [144,145]. This is possibly linked to poor hand hygiene, unclean delivery practices, direct contact with body fluids, and environmental contamination [146]. Few differences seem to exist between reported bacteria of EOS and LOS, contrasting with high-income countries [147]. The occurrence of contaminated environmental samples with resistant germs, notably, MDR Gram-negative bacteria and MRSA *S. aureus*, is alarming and may explain the occurrence of early hospital-acquired infections [148], stressing the need for effective infection control in perinatal care and neonatology.

#### 3.3.3. Diagnostics Challenges

The diagnosis of neonatal sepsis is challenging as the clinical presentation is unspecific, the disease progresses very fast, and warning signs are similar to those of hypoglycemia, hypothermia, or RDS [147,148,149]. White blood cell and differential count have low sensitivities and are of little use due to a broad physiologic variation during the first days of life [149,150]. C-reactive protein (CRP) is the most studied acute phase reactant in neonatal infection. It has proven valuable in high-income settings [151,152,153], where one single CRP value between 8 h and 36 h from admission had a negative predictive value for sepsis of >99% [153]. Using serial CRP measurements to guide antibiotic therapy in neonates has proven safe and practical in developing countries [154]. As sepsis in neonates has fulminant developments, CRP-guided antibiotic treatment may be used to reduce the duration of antibiotic exposure stopping antibiotics early [152,155,156]. Early low CRP, however, cannot exclude sepsis and should not delay pre-emptive initiation of antibiotics [149,156,157] and early elevated CRP cannot be consistently interpreted as sepsis since conditions such as pre-eclampsia and fetal distress may increase it [157].

Blood cultures remain the gold standard to confirm neonatal sepsis. However, the sensitivity of cultures decreases with blood samples below 1 mL, which are very challenging to draw in neonates due to anatomic and technical difficulties [158]. Sensitivity may also be negatively influenced by prior maternal or neonatal antibiotic exposure or low laboratory competencies [148]. Diagnostic challenges are amplified in resource-limited contexts where most deliveries occur at home, and outpatient clinics are rudimentary and overcrowded.

Biomarkers such as CRP, and blood cultures are indeed rarely available in LMIC. Therefore, the diagnosis of neonatal sepsis is mostly presumptive, and unnecessary antibiotic exposure of non-infected newborns increases antibiotic pressure on the environment, favoring the development of bacterial resistance.

#### 3.3.4. Treatment Challenges

Pre-emptive initiation of antibiotic treatment for suspected neonatal sepsis is generally accepted good clinical practice. Laboratory results help narrow antibiotic choice and treatment duration that otherwise remain entirely empirical. The WHO still recommends treatment initiation with ampicillin and gentamicin as first-line choice for neonatal sepsis [159], and for suspected meningitis, when available, a third-generation cephalosporin (ceftriaxone or cefotaxime).

Data on local antimicrobial resistance patterns are often lacking. However, a recent meta-analysis shows alarming results in neonates in sub-Saharan Africa. In this literature review, 89 % of all *E. coli* were resistant to ampicillin. The gentamicin resistance for *E. coli* and *Klebsiella* spp. was 47 and 66%, respectively. The authors also described a high proportion of *E. coli* and *Klebsiella* spp. resistant to ceftriaxone (38% and 49%, respectively). In addition, 50% of *Staphylococcus aureus* were methicillin-resistant [145]. This high level of resistance was confirmed by the BARNARDS network, which collects data on neonatal infections and antibiotic resistance in developing countries [160].

It must be concluded from these reports that the most common bacteria of neonatal infection in LMIC are largely resistant to the present, WHO recommended antibiotic regimen, and that high morbidity and fatality rates from multi-drug resistant micro-organisms have indeed been reported in other studies [161]. While these findings push for the use of second- and third-line antibiotics, their use will invariably lead to new resistances without strict infection control and antibiotic stewardship programs that help shorten unnecessary antibiotic exposure. CRP-guided antibiotic treatment has a high potential to reduce antibiotic exposure and may be cost-effective at the public health level.

## 4. Discussion

Neonatal mortality has been high for so long that it has become a routine. The leading causes and associations of neonatal mortality have been identified: birth-related complications and asphyxia; complications of low birth weight and prematurity, notably hypothermia; and infection. However, these are tightly interlinked, and many neonates will die from mixed conditions. Furthermore, in LMIC undoubtedly even much more, considerable morbidity is poorly identified and quantified, and information on outcome and long-term consequences remain scarce. Therefore, possibly the most critical ingredient for global improvement of neonatal survival would be increased visibility and consideration for human rights and respectful care.

To tackle the burden of neonatal diseases, cost-effective interventions need urgent implementation. In addition, interlinked pathologies require intervention bundles based on specific and detailed data to identify regional bottlenecks and needs. The foundations for improvement lie in better data, deepening the knowledge on morbidities that lead up to the three main causes of death as well as the development of tools and strategies adapted to the local context. Without this knowledge, most improvement strategies will lack the substance for precise, cost-effective, targeted intervention and follow-up.

We identified four clinical domains and other structural broader fields that need priority attention. We described these domains of action according to the leading causes of neonatal mortality. We attributed a specific target for thermal control that, despite being usually reported associated with preterm and LBW death, is not only a problem of the preterm. Without adequate thermal care, even a healthy full-term neonate with adequate birth weight might die from hypothermia, even in tropical climate.

### 4.1. Perinatal Care—Prevention of Asphyxia

Coordinated perinatal quality obstetric and neonatology expertise is needed to reduce perinatal hypoxia and involves antenatal care, skilled intrapartum care, and monitoring fetal wellbeing during delivery. Improving antenatal and intrapartum care has been clearly shown to reduce the fresh stillbirth rate and increase neonatal survival.

Intrapartum care needs uninterrupted continuity with postnatal care by skilled professionals in neonatal resuscitation, thermal control, as well as evaluation and identification of at-risk patients in need of additional follow-up. Systematic immediate evaluation of the newly born neonate by the APGAR score is simple to adopt and provides much more than standardized data to understand early adaptation, it prompts a very early initiation of neonatal monitoring, within the first Golden Minute of life, the essential period when postnatal resuscitation is highly effective. Unfortunately, proven clinical scoring tools, such as the Apgar and Sarnat scores, are often given little appreciation in LMIC for fear of repressive judgment on obstetric performance, where constructive support from healthcare administrations and training may actually lead to their virtuous use in a quality improvement cycle.

Neonatal resuscitation within the first minute is simple and effective in 99% of cases. Its indication is based on two single, easily identifiable parameters, breathing and heart rate. In the 10% of neonates that need some form of it, resuscitation interventions are very standardized, following simple steps that are easy to learn and remember by the acronym T-ABC—thermal control, airways, breathing, and circulation. Basic neonatal resuscitation training requires only a couple of hours of instruction and hands-on training. It is accessible even to healthcare staff with low general education without expensive equipment. It needs an implementation program, with the training of local instructors, staff training and regular refresh.

### 4.2. Thermal Control

Within neonatal resuscitation and throughout neonatal life, thermal stability has been shown to significantly reduce mortality. To maintain thermal balance, it is essential to detect cold stress early, before hypothermia occurs, as it leads to exhaustion of oxygen and glucose even before hampering thermosensitive metabolic processes. Understanding the seriousness of cold stress with a hand-touch evaluation of the extremities can guide low-cost interventions to prevent hypothermia, such as avoiding bathing, rapid drying, skin-to-skin care, and clothing and wrapping. More specifically, for LBW and premature neonates, adapted and robust technical devices will have to be made available for LMIC, including training and maintenance. Updated thermal guidelines stressing the prevention of cold stress, the use of cost-effective technical equipment for those at highest risk and, most importantly, education on thermal care of the newborn are priorities.

### 4.3. Tackling Prematurity

As prematurity prevention has remained unsuccessful so far, it is necessary to apprehend its specificities and prevent complications. Care for the preterm neonate is also multi-disciplinary, starts antenatally with corticosteroids for lung maturation, continues perinatally with safe birth management and requires immediate, anticipated support at birth. Neonatal resuscitation is even more critical than in full-term neonates but remains straightforward, although highly dependent on airways and breathing management that needs smooth extension into supportive care practices. Simple and effective thermal care strategies, such as kangaroo care, also minimize the first cause of death in prematurity, RDS. Hypoxia due to respiratory disease can be avoided in many cases with relatively inexpensive equipment, notably, with blended oxygen and continuous positive airway pressure (CPAP).

Nutrition is vital and should not be delayed but feeding management of the preterm is technical and needs specific training. Breastmilk protects against necrotizing enterocolitis and starvation and needs to start early in preterm neonates too.

Some essential cost-effective technical solutions and strategies are needed, too. Essential, robust, and low-cost equipment must become available globally, such as incubators, phototherapy and adapted CPAP devices [162,163]. The expertise to run and repair these devices is clearly essential and should combine with an appropriate quality control cycle based on reliable data. For a cost-effective support, preterms, therefore, need to be taken care of in reference centers.

### 4.4. Infections

Whether in high- or low-competency centers, neonatal sepsis is highly prevalent in low-income settings and is a leading cause of neonatal mortality, with likely life-long, though still unquantified, disabilities. Education, safe delivery practice, and access to quality care are priorities in the fight against neonatal sepsis. In addition, access to diagnostic tools and microbiological documentation, and the availability of second-and third-line antibiotics is needed. However, without implementing infection control and antibiotic stewardship programs, neonatal mortality and morbidity will continue to soar. Resistances will continue to increase the burden of death and disability globally, making this target of importance for low- and high-income countries alike.

### 4.5. Broader Structural Interventions

It cannot be stressed enough that reliable data is paramount to identify specific bottlenecks and to follow-up corrective interventions. Such data need to include quality of care and user experience parameters. If epidemiological data remains of low quality, un-standardized, and poorly reported, the frightening reality will remain underestimated and poorly understood.

Training healthcare professionals is essential. Retaining and supporting trained staff requires a safe and sustainable work environment with essential functional equipment and an acceptable ratio between patients and healthcare professionals. These conditions are more easily gathered in high-competence centers.

As long as high-risk deliveries occur away from these high-competence centers, a significant part of delivered neonates will need an urgent and safe referral. Neonatal transport is known to worsen the outcome significantly. Today, in LMIC, high-level centers are insufficient in numbers and often neither equipped nor staffed better than lower-level centers. Specific transport equipment and teams are rarely available, further worsening the patient condition during referral, and finally, overwhelming referral centers with dying patients. The resulting high mortality rates undermine the population’s trust, closing a circle of system failures. Regionalization of high-competence centers is essential and requires firstly excellence in low-risk situations, not at least for reputation and education purposes, before expanding expertise into high-risk care. If reference centers only receive desperate cases, their reputation will remain undermined.

#### Rights and Respect for Every Newborn

Although neonates are the future of all societies, it is saddening to realize that their lives remain considered of lesser value. These considerations may originate from the traditional high mortality at birth and inherent desire to avoid investment into such high risk, but also from a legal construct dividing a physiologic continuum into binary rights split by birth. Lower value often comes with lower respect. Neonatal life may well, in the first place, suffer from under-recognition, under-consideration, and lack of respect. Integration of newborn rights into laws and regulations remain an insufficiently met obligation of governments [164], as do clinical protocols and guidelines for delivering respectful and dignified care to newborns. Respectful care is paramount for patient trust. To meet SDG 3.2, the world will need to transform newborn care by acknowledging their legal rights.

## 5. Conclusions

Neonatal health should be considered beyond survival and treated as a high priority from global leaders to local medical staff. The enormous number of deaths hides an even larger number of diseased and impaired, which considerably impacts women, families, and society. Distributive justice would prompt focus on geographical areas with the highest needs: sub-Saharan Africa, South Asia, and conflict zones.

Tackling the three leading overlapping causes of neonatal mortality needs further understanding of the underlying predispositions and pathologies and therefore focused research on causes and cost-effective interventions. To guarantee the success and sustainability of interventions, implementation research focused on education and training in a regionalized network are key factors. 

Increased visibility and investment focused on these areas are urgently needed to reduce the millions of preventable newborn deaths and ensure they reach their full potential. Reducing neonatal mortality and morbidity is much more than investing in neonates; it is improving adult health and constructing and perpetuating a stable society and a thriving economy.

Clearly interdependent with maternal health, neonatal health has its specific, well-defined, and comprehensive targets, particularly in terms of pathologies. In health policies, neonates should be visible, and interventions must aim to offer sustainable high-quality care. To make that possible, we believe there is a paramount need to change the mindset, acknowledging and addressing the fundamental rights of this vulnerable population, the neonates.

## Data Availability

Not applicable.

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
