# Peer review of "2.5 Million Annual Deaths—Are Neonates in Low- and Middle-Income Countries Too Small to Be Seen? A Bottom-Up Overview on Neonatal Morbi-Mortality"

_tropicalmed, 2022, doi:10.3390/tropicalmed7050064_

Round 1

Reviewer 1 Report

I appreciate the sincere effort to describe neonatal morbidity and mortality in lower-income countries. This article should be written as a review/opinion piece rather than an expert review, which is reader-friendly. Introduction, as well as Methods, should be edited for brevity. It is a review, that doesn't warrant a result section, a well-written discussion should suffice.

Major comment: It would be helpful if you used the global data from WHO and divided it into different geographical areas(separating south Asia and Africa) and then commented on the mortality and the causes of mortality. Though it is a long write-up, it doesn't add much to the literature. A lot of hard work is being done in a lot of LMIC countries to provide care at the grass root level, to prevent neonatal mortality(including prioritizing the health budget for neonatal care). The manuscript does nothing to mention or explain either the WHO-led initiates or local/regional initiatives.

Even though the manuscript is written with good intentions but parts of the manuscript are based on opinion and feel insensitive. Please consider re-writing" Althous this definition... no skilled birth attendant is available"  " "Traditionally delaying naming.... early death" " The neonatal period... low recognition"..." Unfortunately immediate newborn intervention is missed "repressive judgment on obstetric performance"  " LMIC hospital's reputation is often blemished by high mortality rates" 

Author Response

  • We thank reviewer 1 for the thorough reading of our work. Still, We believe he/she has misunderstood the scope of our work that wants to report a bottom-up view of an undoubted lack of improvement in neonatal mortality and morbidity. We acknowledge that this may result from an insufficiently specific methods section. As the editor has requested more details in the methodology, we clarified the section and changed the title to specify the methodology: an overview ( Grant et al. Health info Libraries J. 2009 91-108). We understand the reviewer's point of view, but the proposed massive changes would entirely change the scope of the paper too.
  • Many papers follow already the reviewer's suggestions, but by following this classical approach we could not provide an original bottom-up realist review that highlights a lesser-known angle. We edited the methodology section to clarify our goals to account for some of the critiques. We also emphasised that experts with numerous years of experience in their field of expertise (in high- and low-resource settings) developed each section. In our very clinical approach, we wish to link frequent problems from the field with practical clinical interventions to reduce neonatal morbi-mortality. 
  • We acknowledge that this overview may pinpoint some politically at least controversial subjects. However, the current disappointing progress in neonatal care in LMIC warrants, in our opinion, a different, non-political approach and angle of view. Therefore, we considered though some linguistic adaptations during English revision.      

Reviewer 2 Report

This review provides a good overview of neonatal death causes and interventions that could reduce mortality. The authors might consider the following minor comment for improvement 

1-Although the authors have provided with detail introduction information, the rationale for this review is missing, please consider adding it 
2-The references numbering in the text should be revised 
3-The text format is not uniform, please revise it 

Author Response

We thank the reviewer for the constructive and relevant comments. Please find the answers point by point below:

  1. We believe some of the rationale was given in the introduction, but we have now explicitly added the rationale into methods. Indeed our rationale was to take the focus from policy to the clinical aspects useful and understandable by the clinician in the field. We do believe that linking the presented bottom-up approach to the top-down view of the policymakers would ultimately lead to progress.
  2. References have been checked and corrected - with thanks!
  3. 3. Format has been adjusted - with thanks!

Reviewer 3 Report

The authors have produced a very comprehensive and extensive review of the problems of the neonate including morbidity and mortality together with the underlying causes. Resources and education seem to remain outstanding issues. The review is timely and makes its points cogently.  The English is generally satisfactory, however, there are a few issues needing attention: for instance on Page 9 "banalisation of hypothermia." I am not sure what this means. I think the authors could find a more appropriate term. On Page 12 "several transversal fields" requires a more understandable sentence. 

On Page 13, 4.2 Thermal Control. "shown to vary significantly reduce mortality" needs fixing. Finally, on Page 14 the heading "Transversal interventions" is obscure. Do the authors mean "broader interventions"?

In the discussion, would maternal education with female empowerment be a helpful goal to reduce neonatal morbidity and mortality? 

Author Response

We thank the reviewer for these constructive and relevant comments!

  • Concerning the expressions:  "banalization", we meant "the process of becoming or making something banal", a word that figures in English dictionaries. However, we have now changed it to 'trivialization' as a more common word.
  • We have now eliminated the word 'vary' that had been forgotten during reformulations and did not make sense. We have also changed 'transversal interventions' into the suggested 'broader interventions'.
  • We have considered adding this point during the process. Still, after discussing with our team, we decided not to include it to avoid deviating the focus from the leading technical neonatal problems. Nevertheless, we can add this point to the discussion if the editor wishes.     

Reviewer 4 Report

The manuscript has been reviewed. The author conducted a pragmatic review on neonatal morbi-mortality in low- and middle-income countries. The manuscript gave a very detailed description about the reasons why the review is necessary. It also described the definitions and results about the morbidity and mortality of neonates in LMIC. From this manuscript, we could see many possible causes that lead to morbidity and mortality of neonates. Even though the author has given a great effort to complete this manuscript, there are some problems that require further explanation and refine.

  1. In the Abstract part, there is no section (3), which I suppose is the result part.
  2. The description in the Materials and Methods part is too simple. Since this manuscript is a review article, it would be better to explain what and how literatures are selected to review with more detail.
  3. The numbers and position of the list numbers in Results part require further checkout.

Despite the above small problems, the manuscript is well written.

Author Response

Thank you for the constructive and relevant comments. Please find the answers point by point below:

  1. The numbers have been corrected. We have chosen to put results and discussion together for the abstract only, as the results are more qualitative than quantitative and would have expanded the section beyond the available space unless oversimplified. 
  2. We acknowledge that the methodology section needed an edition, we did it so making specific our rationale, the design and the process of this review. Indeed it was not a systematic process, so we clarified the section and changed the title to specify the methodology- overview (Rycroft-Malone et al. Implementation Science 2012, 7:33; Grant et al. Health Info Libraries J 2009 91-108). We deliberately chose not to do a systematic review in order to show a different approach "bottom-up" based on clinical challenges and neonatal physiology. We hope that by adapting the methods and title using the term 'overview' we can refocus the scope for the reader. 
  3. This has been corrected now - thank you!